

# Antarctic Bedmap data: FAIR sharing of 60 years of ice bed, surface and thickness data

Alice C. Frémand [1*], Peter Fretwell [1*], Julien Bodart [2,1*], Hamish D. Pritchard [1], Alan Aitken [3], Jonathan L. Bamber [4,50], Robin Bell [5], Cesido Bianchi [6], Robert G. Bingham [2], Donald D. Blankenship [7], Gino Casassa [8], Ginny Catania [7], Knut Christianson [9], Howard Conway [9], Hugh F.J. Corr [1], Xiangbin Cui [10], Daniel Damaske [11], Volkmar Damm [12], Reinhard Drews [13], Graeme Eagles [14], Olaf Eisen [14,51], Hannes Eisermann [14], Fausto Ferraccioli [15,1], Elena Field [1], René Forsberg [16], Steven Franke [14], Shuji Fujita [17], Yonggyu Gim [18], Vikram Goel [19], Siva Prasad Gogineni [20], Jamin Greenbaum [21,7], Benjamin Hills [22], † deceased Richard C.A. Hindmarsh [1], Per Holmlund [23], Nicholas Holschuh [24], John W. Holt [25], Angelika Humbert [14,51], Robert W. Jacobel [26], Daniela Jansen [14], Adrian Jenkins [27], Wilfried Jokat [14,51], Tom Jordan [1], Edward King [1], Jack Kohler [28], William Krabill [29], Kirsty A. Langley [30], Joohan Lee [31], German Leitchenkov [32], Carlton Leuschen [33], Bruce Luyendyk [34], Joseph MacGregor [35], Emma MacKie [36,52], Kenichi Matsuoka [37], Mathieu Morlighem [38], † deceased Jérémie Mouginot [39,53], Frank O. Nitsche [5], Yoshifumi Nogi [17], Ole A. Nost [37], John Paden [33], Frank Pattyn [40], Sergey V. Popov [41], Mette Riger-Kusk [42], Eric Rignot [39,54,18], David M. Rippin [43], Andrés Rivera [44], Jason Roberts [45,55], Neil Ross [46], Anotonia Ruppel [12], Dustin M. Schroeder [36,56], Martin J. Siegert [47], Andrew M. Smith [1], Daniel Steinhage [14], Michael Studinger [48], Bo Sun [10], † deceased Ignazio Tabacco [6], Kirsty Tinto [5], Stefano Urbini [6], David Vaughan [1], Brian C. Welch [26], Douglas S. Wilson [49], Duncan A. Young [7], Achille Zirizzotti [6]

[1]British Antarctic Survey, Cambridge, UK.
[2]School of GeoSciences, University of Edinburgh, Edinburgh, UK.
[3]Centre for Exploration Targeting, University of Western Australia, Crawley, Australia
[4]School of Geographical Sciences, University of Bristol, UK
[5]Lamont-Doherty Earth Observatory of Columbia University, Palisades, USA
[6]Istituto Nazionale di Geofisica e Vulcanologia, Rome, Italy
[7]Institute for Geophysics, University of Texas at Austin, USA
[8]Centro de Estudios Cientificos, Santiago, Chile
[9]Earth and Space Sciences, University of Washington, Seattle, USA
[10]Polar Research Institute of China, Shanghai, China
[11]Center for Marine Environmental Sciences, University Bremen, Bremen, Germany
[12]Federal Institute for Geosciences and Natural Resources, Hannover, Germany
[13]Tübingen University, Department of Geosciences, Tübingen, Germany
[14]Alfred Wegener Institute, Helmholtz Centre for Polar and Marine Research, Bremerhaven, Germany
[15]Istituto Nazionale di Oceanografia e di Geofisica Sperimentale, Trieste, Italy
[16]DTU Space, Lyngby, Denmark
[17]National Institute of Polar Research, Tokyo, Japan
[18]Jet Propulsion Laboratory, California Institute of Technology, Pasadena, USA
[19]National Centre for Polar & Ocean Research (NCPOR), Ministry of Earth Sciences, Vasco-da Gama, Goa - 403804, India
[20]University of Alabama, Tuscaloosa, AL 35487, USA
[21]Scripps Institution of Oceanography, La Jolla, USA
[22]Department of Earth and Space Sciences, University of Washington, Seattle, USA





[23]Stockholm University, Stockholm, Sweden

[24]Amherst College, Amherst, USA

[25]University of Arizona, Tucson, USA

[26]St. Olaf College, Northfield, MN 55057, USA

[27]Northumbria University, Newcastle, UK

[28]Norwegian Polar Institute, Fram Centre, Tromsø, Norway

[29]NASA Wallops Flight Facility, Virginia, USA

[30]Asiaq, Greenland Survey, Nuuk, Greenland

[31]Korean Polar Research Institute, Incheon, South Korea

[32]Institute for Geology and Mineral Resources of the World Ocean, St. Petersburg, Russia

[33]Centre for Remote Sensing of Ice Sheets, University of Kansas, Lawrence, USA

[34]Earth Research Institute, University of California in Santa Barbara, USA

[35]Cryospheric Sciences Lab, NASA Goddard Space Flight Center, Greenbelt, Maryland, USA

[36]Department of Geophysics, Stanford University, Stanford, CA, USA

[37]Norwegian Polar Institute, Tromso, Norway

[38]Department of Earth Sciences, Dartmouth College, Hanover, USA

[39]Department of Earth System Science, University of California Irvine, Irvine CA, USA

[40]Laboratoire de Glaciologie, Université Libre de Bruxelles, Brussels, Belgium

[41]Polar Marine Geosurvey Expedition, St. Petersburg, Russia

[42]Western Norway University of Applied Sciences, Bergen, Norway

[43]Department of Environment and Geography, University of York, York, UK

[44]Departamento de Geografía, Universidad de Chile, Santiago, Chile

[45]Australian Antarctic Program Partnership, Institute for Marine & Antarctic Studies, University of Tasmania, Hobart, Australia

[46]School of Geography, Politics and Sociology, Newcastle University, Newcastle-upon-Tyne, UK.

[47]Grantham Institute and Department of Earth Science and Engineering, Imperial College London, London, UK

[48]NASA Goddard Space Flight Center, Greenbelt, USA

[49]Marine Science Institute, University of California Santa Barbara, USA

[50]Department of Aerospace and Geodesy, Technical University of Munich, Germany

[51]Department of Geoscience, University of Bremen, Bremen, Germany

[52]Department of Geological Sciences, University of Florida, USA

[53]University of Grenoble Alpes, CNRS, IRD, Grenoble INP, IGE, Grenoble, France

[54]Department of Civil and Environmental Engineering, University of California Irvine, Irvine CA, USA

[55]Australian Antarctic Division, Kingston, Australia

[56]Department of Electrical Engineering, Stanford University, Stanford, CA, USA

*These authors contributed equally to this work.

*Correspondence to*: Alice C. Frémand (almand@bas.ac.uk) and Peter Fretwell (ptf@bas.ac.uk).



**Abstract.** Over the past 60 years, scientists have strived to understand the past, present and future of the Antarctic Ice Sheet. One of the key components of this research has been the mapping of Antarctic bed topography and ice thickness parameters that are crucial for modelling ice flow and hence for predicting future ice loss and ensuing sea level rise. Supported by the Scientific Committee on Antarctic Research (SCAR), the Bedmap3 Action Group aims not only to produce new gridded maps of ice thickness and bed topography for the international scientific community, but also to standardize and make available all

the geophysical survey data points used in producing the Bedmap gridded products. Here, we document the survey data used in the latest iteration, Bedmap3, incorporating and adding to all of the datasets previously used for Bedmap1 and Bedmap2, including ice-bed, surface and thickness point data from all Antarctic geophysical campaigns since the 1950s. More specifically, we describe the processes used to standardize and make these and future survey and gridded datasets accessible under the 'Findable, Accessible, Interoperable and Reusable' (FAIR) data principles. With the goals to make the gridding

process reproducible and to allow scientists to re-use the data freely for their own analysis, we introduce the new SCAR Bedmap Data Portal (bedmap.scar.org, last access: 18 October 2022) created to provide unprecedented open access to these important datasets, through a user-friendly webmap interface. We believe that this data release will be a valuable asset to Antarctic research and will greatly extend the life cycle of the data held within it. Data are available from the UK Polar Data Centre: https://data.bas.ac.uk.

**1 Introduction**

Detailed and extensive information on ice thickness and bed topography is needed to reconstruct the geological and geomorphic history of Antarctica, and to model ice flow in order to predict the ice sheet's future contribution to sea level rise (Fretwell et al., 2013; DeConto and Pollard, 2016; Scambos et al., 2017; The IMBIE team, 2018; Rignot et al., 2019; Morlighem et al., 2020; DeConto et al., 2021; Fox-Kemper et al., 2021). This information has primarily been gathered using

ground-based or airborne radio-echo sounding (RES) and seismic surveys conducted by over 50 institutions under multiple national programmes across Antarctica over the last 60 years. However, up until now, these survey datasets had not been held centrally nor been standardized, thus limiting their accessibility to the wider Antarctic community. Consequently, previous attempts to map the ice sheet on the continental scale, such as Bedmap 1 (Lythe et al, 2001), Bedmap2 (Fretwell et al., 2013) and Bedmachine Antarctica (Morlighem et al., 2020), have had to first find data, gain permissions, download, clean and

standardize hundreds of datasets from survey campaigns of many different sources, before finally constructing the grids. Theses constraints have led to only a limited number of gridded products being made, often years apart and with a long lag after the surveys have been completed. Given the rapidity of change affecting large parts of the Antarctic Peninsula and threatening the stability of the West Antarctic Ice Sheet, and the urgency in predicting future ice loss (e.g. Mouginot et al., 2014; Golledge et al., 2015; DeConto & Pollard 2016; Gardner et al., 2018; Seroussi et al., 2020; Levermann et al., 2020), it

is essential for these data to be freely available to the international community.



Supported by the Scientific Committee of Antarctic Research (SCAR) Bedmap3 Action Group, this paper presents the release of all of the underlying ice bed, surface and thickness survey data points that have been used in the previous and upcoming versions of Bedmap gridded products (Bedmap1, Bedmap2 and Bedmap3). We discuss the standardisation of the data following the Findable, Accessible, Interoperable and Reusable (FAIR) data principles (Wilkinson, 2016) and the use of consistent data formats and attributes, as agreed by the international community through the Bedmap project. Additionally, we introduce the SCAR Bedmap Data Portal (bedmap.scar.org, last access: 18 October 2022) which offers the ability to search individual datasets within one stand-alone map-based platform and increases the discoverability and accessibility of the data. Our aim is to make the gridding process as reproducible as possible by making the source survey data fully standardised, openly available and easily accessible through one portal. It is expected that the data presented in this paper will facilitate the creation of a range of new gridded products at different spatial resolutions, enable the application of emerging techniques such as machine learning and geostatistical techniques to fill gaps between direct measurements, and provide a common data sharing baseline for future geophysical surveying of Antarctica. A follow-up publication to this paper will introduce the new gridded products from Bedmap3.

Section 2 of this paper discusses the background and evolution of past surveying of Antarctica using geophysical techniques. Section 3 presents how the source data have been standardised. Section 4 details how the data are published following the FAIR data principles.

## 2. Background: evolution of the Bedmap products

### 2.1 1950 – 1980: First geophysical measurements of ice thickness in Antarctica

Prior to the start of radio echo-sounding (RES) measurements over Antarctica, ice thickness was primarily obtained from seismic techniques (Schroeder et al., 2020). RES was developed in the 1950s after studying the transparency of ice to specific radio frequencies and the realisation of its potential for glaciological research by Armory Waite and Stanley Evans (Turchetti et al., 2008). After several years of developments and tests, the first long-range airborne radio-echo sounding of the Antarctic Ice Sheet was undertaken by the Scott Polar Research Institute (SPRI), with support from the United States National Science Foundation and the Technical University of Denmark in the late 1960s (Robin et al., 1970). By 1975, the elevation data from the 1971-1975 Antarctic field seasons were compiled into a series of topographic maps of Antarctica (Drewry, 1975). These became the first comprehensive topographic maps of the Antarctic continent and would lead to more sophisticated compilation grids in the following years.

### 2.2 1980 – 1990: First compilation efforts to map Antarctica

By 1983, around 50% of the Antarctic ice sheet had at least some airborne RES survey measurements (i.e. within a 50 to 100 km square grid cell) (Drewry et al., 1982) and the first compilation bed elevation map was published. Sheets 3 and 4 in

the SPRI Glaciology and Geophysical Folio Series (Drewry, 1983) rapidly became a reference for bedrock surface and ice thickness for Antarctica. The grid contours of bed elevation were drawn from ice thickness data collected on sparse surface

traverses and by airborne surveys over the entire continent, using state-of-the-art digital mapping techniques, although in many areas survey lines were separated by hundreds of km. (Lythe et al., 2000).

## 2.3 1990 - 2020: The Bedmap era

In the mid-1990s, considerable advances in radar data acquisition and development of modern Global Navigation Satellite Systems (GNSS) led to substantial improvements in the coverage and accuracy of the data collected. Indeed, until then the

positioning was often inferred using the "unaided inertial navigation" technique which often had substantial positioning errors (Schroeder et al. 2020).

In 1996, the first BEDMAP consortium group (here termed Bedmap1), was set up under the joint sponsorship of the European Ice Sheet Modelling Initiative (EISMINT) and SCAR. It led to the publication of the first Bedmap products: a printed map published in 2000 (Lythe et al., 2000) and its associate digital version in 2001 (Lythe et al, 2001). For more than a decade,

Bedmap1 played a crucial role in providing large-scale boundary conditions of the Antarctic Ice Sheet for observational and modelling applications (e.g. Pollard and DeConto, 2009; Shepherd et al., 2012). The gridded map contained ice thickness data from direct measurements, including ground-based and airborne RES but also from seismic and gravimetric measurements (Lythe et al, 2001). Although pioneering, this first gridded product had a relatively low resolution of 5-km and suffered from large data gaps particularly over East Antarctica, which resulted in low-confidence values in those areas (see Figure 1a).

Motivated by a wealth of newly acquired data over Antarctica and improved Geographic Information System techniques, the second version of Bedmap was published in 2013 (Fretwell et al., 2013). The Bedmap2 product was composed of several grids including ice bed, surface and thickness data for Antarctica and their associated uncertainties, in addition to several masks (e.g. continental ice-edge, grounding line, ice-shelf extent, etc.) useful for ice-sheet modelling. This compilation included 25 million measurements, an order of magnitude more than were used in Bedmap1. This time, the ice thickness, bed and surface

elevation grids were provided at a uniform 1-km spacing, but still with a native interpolation resolution of 5-km in order to satisfy the data providers' conditions for use (Fretwell et al., 2013).

Since 2012, new RES datasets have been collected across Antarctica, with a particular focus on the 'poles of ignorance' identified in Bedmap2 (Pritchard, 2014), thus filling known data gaps in key areas of East Antarctica (see Sect. 3.1). In addition, new hybrid compilation efforts such as BedMachine Antarctica have used a combined modelling-observation

approach, including a mass conservation method, to generate an improved bed topography and ice thickness in data-deficient areas of the Antarctic coastline (Morlighem et al., 2020).

## 2.4 2020 - Present: General Approach for Bedmap3

In 2020, the SCAR Bedmap3 Action Group was tasked to produce an updated version of the Bedmap gridded products and to improve the accessibility of the underlying survey datasets of Antarctic ice thickness and bed topography (see Figure

1a-c) through standardisation and dissemination of the data via a new SCAR Bedmap data Portal. This will serve as a common endpoint to discover and interact with all underlying Bedmap data.

The Bedmap3 gridded products will be constructed using a similar process to Bedmap2 but will offer a significant improvement in survey-data coverage, along with a newly updated grounding line, updated altimetry-derived surface topography, and updated ice extent and bathymetry. Each iteration of Bedmap contains large survey-data additions that have

increased the accuracy of the gridded products. In total, Bedmap3 contains 77 million points and thus includes twice the number of new data points available to Bedmap2 (Figure 1d, Table 1 and S2).

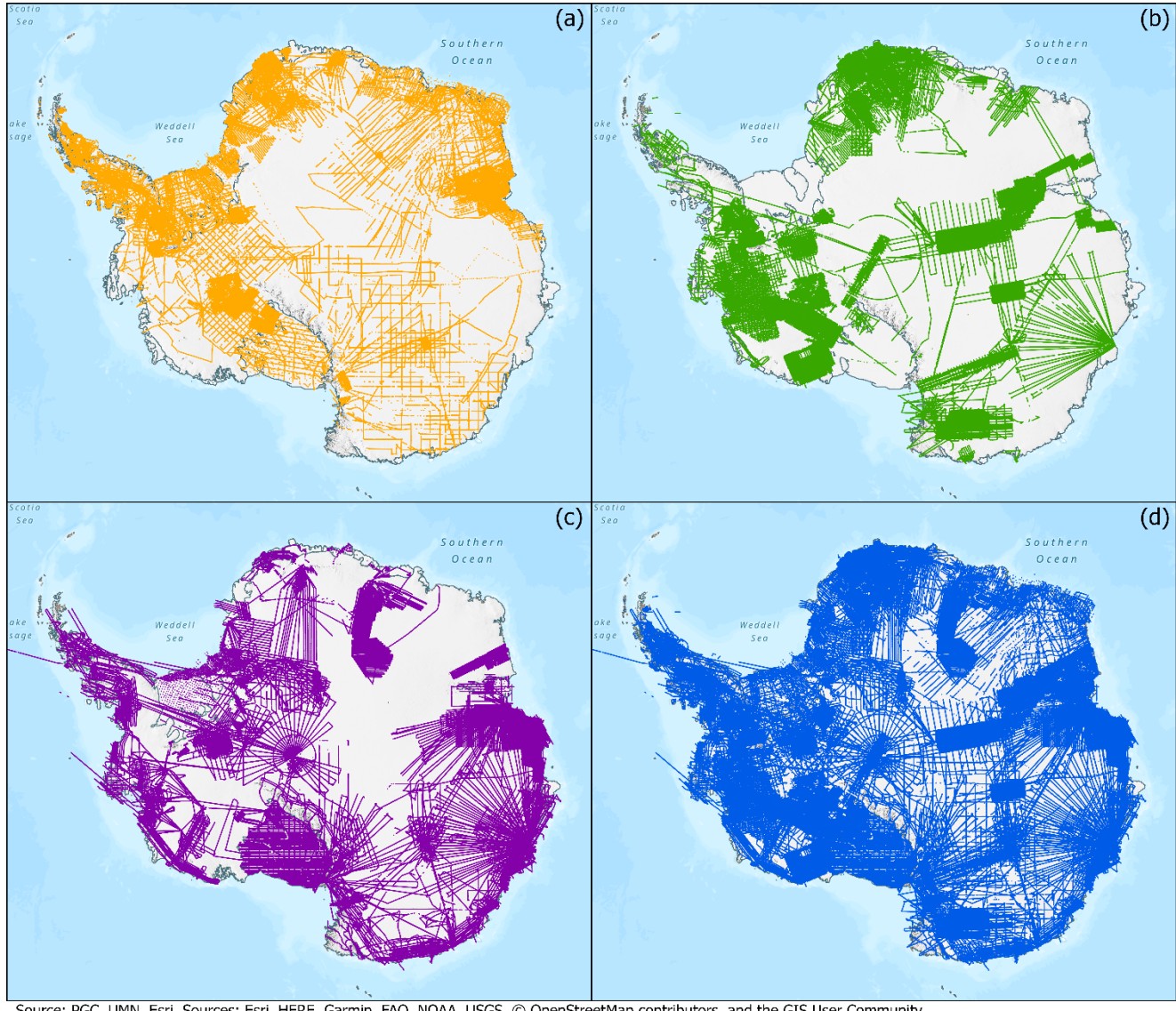

Source: PGC, UMN, Esri, Sources: Esri, HERE, Garmin, FAO, NOAA, USGS, © OpenStreetMap contributors, and the GIS User Community

**Figure 1: Data coverage for the three generations of Bedmap products. (a) Data coverage for Bedmap1. (b) Additional data coverage for Bedmap2. (c) Additional data coverage for Bedmap3. (d) Total combined coverage now available.**



## 3. Source data, standardisation and pre-processing

### 3.1 Ice thickness, surface and bed elevation data

The primary source data consist of survey point measurements of ice thickness, bed elevation and surface elevation, which principally comes from airborne radar surveys and seismic soundings, and to a smaller extent from ground-based radar surveys. We present here the data compiled within each version of Bedmap.

Bedmap1 source data (1950s-1990s) often lack the campaign metadata available for more modern datasets, and so we present these as a single dataset. In total, the data standardised for Bedmap1 consists of almost 2 million points from 127 individual campaigns (Table 1). While the data coverage is substantial, especially over West Antarctica and the Antarctic Peninsula (Fig. 1a), the distance between individual flight lines and soundings is much larger than those of the Bedmap2 and Bedmap3 data. In addition, though efforts continue to leverage modern data to improve the geometric, positioning, and radiometric calibration for this archival data, the spatial accuracy of the survey data is poorer due to the use of older navigation techniques prior to the GNSS era (see Schroeder et al., 2019; Schroeder et al., 2021).

Additional Bedmap2 source data were acquired from 2000 to 2012 by 65 new surveys that contributed a further 25 million points (Table S1), filling major gaps over West Antarctica's fast-flowing ice streams such as Pine Island (Vaughan et al., 2006) and Thwaites (Holt et al., 2006) glaciers, as well as over East Antarctica's Gamburtsev Subglacial Mountains (Sun et al., 2009; Bell et al., 2011; Ferraccioli et al., 2011) and Wilkes Subglacial Basin (Frederick et al., 2016) (Fig. 1b).

Further new data available to Bedmap3 come from 84 new surveys by 15 data providers, representing an additional 50 million data points and 1.5 million line-kilometres of measurements (Table S2). These latest data have filled major gaps, particularly in key sector of East Antarctica, including the South Pole (Jordan et al., 2018) and Pensacola basin (Paxman et al., 2019), Dronning Maud Land, Recovery Glacier (Forsberg, 2017) and Dome Fuji (Eagles et al., 2018; Karlsson et al., 2018), and Princess Elizabeth Land (Cui et al., 2020; Popov, 2020). Additional data covering glacier troughs and floating ice shelves give insights into previously under-sampled sectors, such as over the Antarctic Peninsula, West Antarctic coastlines or over the TransAntarctic Mountains as part of NASA Operation IceBridge (MacGregor et al., 2021).

**Table 1: Comparison of data campaigns and coverage of the different Bedmap generations.**

| Bedmap Version | Cumulative campaigns | Cumulative data points | Cumulative data set volume |
|---|---|---|---|
| Bedmap1 | 127 | 2 million | 213 MB |
| Bedmap2 | 192 | 27 million | 6.3 GB |
| Bedmap3 | 276 | 77 million | 21 GB |



## 3.2 Standardisation

Due to the large number of data providers and lack of common protocols, the data received as part of Bedmap data calls came in various forms, including text files, CSV, ASCII, or Excel files. To ensure long-term accessibility, all submitted data files were standardised based upon a template agreed by the SCAR action group and converted to a specific Comma
Separated Value (CSV) format. Open and easy-to-use, this format has been widely used in the scientific community and is well-suited to store tabular data.

As an ASCII-delimited file, the CSV format allows long-term preservation of the data thanks to its very simple structure. This format is often recommended (e.g., https://www.gov.uk/government/publications/recommended-open-standards-for-government/tabular-data-standard, last access: 29 July 2022) but no strict definition exists. One common
definition is the RFC 4180 definition (https://www.ietf.org/rfc/rfc4180.txt, last accessed: 29 July 2022 ) which describes the CSV format as tabular data with 0 to 1 header rows, followed by the same number of field separated by commas. With only one row header, the metadata allowed by this definition is extremely poor. To add information, a solution is to repeatedly add the metadata to each data row, but at the cost of greatly increasing the file size. That is why, the CSV on the Web (CSVW) standard developed by the W3C working group (https://www.w3.org/TR/tabular-data-
primer/"https://www.w3.org/TR/tabular-data-primer/, last accessed: 29 July 2022) or NCCSV, a NetCDF-compatible ASCII CSV file (https://coastwatch.pfeg.noaa.gov/erddap/download/NCCSV.html, last accessed: 29 July 2022) developed by NOAA recommend adding the metadata or notes in a separated JSON or CSV file. Although the metadata as described by the CSVW or NCCSV recommendations are excellent in terms of machine-readability, the metadata are hidden in a complicated structure that compromises human-readability.  For this reason, we used an extended version of CSV that
purposely does not follow the RFC 4180 definition but provides the possibility to add metadata in the data file itself. Different definitions of such format exist such as the geoCSV format developed within the GeoWS project (http://geows.ds.iris.edu/documents/GeoCSV.pdf, last accessed: 29 July 2022) or the extCSV recommended by the World Ozone and Ultraviolet Radiation Data Centre (https://woudc.org/about/formats.php, last accessed: 29 July 2022).

The format used for the Bedmap Data Portal follows most of the geoCSV recommendation, headers are compliant with
the CF convention (https://cfconventions.org/, last accessed: 29 July 2022) and include recommended attributes from the Attribute Convention for Data Discovery (ACDD, https://wiki.esipfed.org/Attribute_Convention_for_Data_Discovery_1-3, last accessed: 29 July 2022). As part of the standardisation, a specific header and structure consisting of identical variable names in a strict order for all the ice thickness data was developed in order to simplify access, particularly for programming purposes. The format consists of (i) an extended header section where (a) each line is introduced by a comment ("#") character,
(b) each line contains a single header item, (c) the colon character (":") is used as the key/value separator, (d) units are in parentheses, (e) attributes preferably use a common vocabulary such as the CF convention and includes attributes from the ACDD; (ii) a header row composed of the column name following the CF convention and units in parentheses; and finally (iii) the data using comma as the separator. The extended header consists of general information regarding each campaign such as



the year, the name of the main investigator, funding and processing details. The complete list and order of the attributes and
variables is given in Table 2.

The developed format is machine readable making the conversion of the files to CSVW or NCCSV standards straightforward if necessary.

**Table 2. List of variable and attribute names provided in the CSV files** To guarantee the machine readibility of the variable names,
the use of special characters was avoided. Conventions include the CF convention (https://cfconventions.org/, last accessed: 29 July 2022)
and recommended attributes from the Attribute Convention for Data Discovery (ACDD, https://wiki.esipfed.org/Attribute_Convention_for_Data_Discovery_1-3, last accessed: 29 July 2022)

| Variable or attribute name | Unit | Details | Convention |
|---|---|---|---|
| **Extended header information** | | | |
| project | | Name of the project or campaign name | ACDD |
| time_coverage_start | Year | Start time of acquisition | ACDD |
| time_coverage_end | Year | End time of acquisition | ACDD |
| creator_name | | Name of contact person or institute responsible for the creation of the dataset | ACDD |
| institution | | The name of the institution principally responsible for originating this data | ACDD, CF |
| acknowledgement | | Name of the funding agency | ACDD |
| source | | Digital Object Identifier for where the original data is deposited | ACDD, CF |
| references | | References pointing to the main publication or discussion of the dataset | ACDD |
| platform | | Type of platform used for the survey: ground-based radar, airborne radar or seismic | CF |
| instrument | | Name of the instrument system used for the acquisition | ACDD |
| history | | Acquisition or processing lineage information | ACDD |
| electromagnetic_wave_speed_in_ice | meters/microseconds (m/μs) | Electromagnetic wave speed in ice | |
| firn_correction | Meters (m) | Firn correction | |
| centre_frequency | MegaHertz (MHz) | Centre frequency | |
| comment | | Comment section used to give the Bedmap version | ACDD, CF |
| metadata_link | | Link to the Bedmap Digital Object Identifier | ACDD |
| license | | URL of the license used | ACDD |
| conventions | | Name of the conventions used | ACDD |
| **Variable names** | | | |
| trajectory_id | | Line or Flight ID | CF |
| trace_number | | Trace number from the specific line given in Line_ID | |
| longitude | decimal degrees (east) | Longitude (WGS84 EPSG: 4326) | CF |



| latitude | decimal degrees (north) | Latitude (WGS84 EPSG: 4326) | CF |
|---|---|---|---|
| date | YYYY-MM-DD | Date following ISO 8601 format: YYYY: year, MM: month, DD: day | |
| time_UTC | HH:MM:SS | UTC time following ISO 8601 format: HH: hours, MM: minutes, SS: seconds | |
| surface_altitude | Meters (m) | Surface elevation or altitude (referenced to WGS84) | CF |
| land_ice_thickness | Meters (m) | Ice thickness | CF |
| bedrock_altitude | Meters (m) | Bed elevation or altitude (referenced to WGS84) | CF |
| two_way_travel_time | Seconds (s) | Two-way travel time | |
| aircraft_altitude | Meters (m) | Aircraft elevation or altitude when applicable (referenced to WGS84) | |
| along_track_distance | Meters (m) | Distance in the along-track direction | |

### 3.3 Summarised point data

Following standardisation, the CSV data (see Section 3.1) were converted to shapefile and geopackage lines and points. Lines were calculated automatically from the point data and split each time a gap of more than 5 km between two data points was found. For Bedmpa1, due to the difficulty in extracting the different campaign data and the sparsity of points, it was not possible to convert Bedmap1 data to shapefile or geopackage lines: only the Bedmap1 shapefile points are provided as part of this data release.

The spatial distribution of the full-resolution survey point data is extremely heterogeneous with, for example, dense, metre-scale sampling along modern flight-lines that are often separated across-track by kilometres to hundreds of kilometres, and this heterogeneity varies between campaigns and data providers. Such an uneven data distribution can cause gridding algorithms to be overly weighted to those areas with the highest sampling frequency, to the detriment of adjacent areas with valid data but sparser sampling. To reduce the impact of data density on gridding, the statistically-summarised

shapefile/geopackage point dataset (centred on a continent-wide 500 m x 500 m grid) reports the average values of the full-resolution survey data plus information on their distribution (Table 3). These summary statistics enable assesment of the confidence in the averaged data values and the variability of the measurements within each cell (e.g., bed roughness). Figure 2 gives an insight to the mean values of ice thickness, bed and surface elevation as well as the number of points per cells used for the calculation.




Source: PGC, UMN, Esri, sources: Esri, HERE, Garmin, FAO, NOAA, USGS, © OpenStreetMap contributions, and the GIS User Community

**Figure 2 Statistically-summarised data points.** (a) Mean surface elevation in meters over Antarctica. (b) Mean bed elevation in meters over Antarctica. (c) Mean ice thickness in meters over Antarctica. (d) Number of points per cell used for the calculation of ice thickness. All elevation values in (a-b) are given with reference to the WGS84 ellipsoid.





**Table 3. List of summary statistics calculated for each shapefile point. For each variable, we provide its short name, long name and associated unit when applicable. These statistics are calculated for each point of the shapefile points file.**

| Short name | Long name | Units |
|---|---|---|
| Cnt_bed | Number of points for bed elevation | - |
| Cnt_surf | Number of points for surface elevation | - |
| Cnt_thick | Number of points for ice thickness | - |
| IQR_bed | Interquartile range for bed elevation points | Meters |
| IQR_surf | Interquartile range for surface elevation points | Meters |
| IQR_thick | Interquartile range for ice thickness points | Meters |
| Max_bed | Maximum value of bed elevation | Meters |
| Max_surf | Maximum value of surface elevation | Meters |
| Max_thick | Maximum value of ice thickness | Meters |
| Mean_bed | Mean value of bed elevation | Meters |
| Mean_dist | Mean distance between cell centre and points | Meters |
| Mean_surf | Mean value of surface elevation | Meters |
| Mean_thick | Mean value of ice thickness | Meters |
| Med_bed | Median value of bed elevation | Meters |
| Med_surf | Median value of surface elevation | Meters |
| Med_thick | Median value of ice thickness | Meters |
| Min_bed | Minimum value of bed elevation | Meters |
| Min_surf | Minimum value of surface elevation | Meters |
| Min_thick | Minimum value of ice thickness | Meters |
| SD_bed | Standard deviation of bed elevation | Meters |
| SD_surf | Standard deviation of surface elevation | Meters |
| SD_thick | Standard deviation of ice thickness | Meters |
| STE_bed | Standard error of bed elevation | Meters |
| STE_surf | Standard error of surface elevation | Meters |
| STE_thick | Standard error of ice thickness | Meters |



### 3.4 Quality control and limitations

The purpose of this data release is to include all possible data collected over the last 60 years without discriminating on the quality of the data. Data has been directly compiled from the data providers, with only minimal quality checks: all non-value data were converted to –9999, including any negative ice thickness values and any points with clear outliers. We checked the minimum and maximum values of each field to ensure the data are in a reasonable range and calculated mean and standard deviation on each dataset to identify potential issues. For example, if longitude/latitude values did not fall within the expected -180 to 180 or -50 to -90 degrees range respectively, the entire row was removed. When no ice thickness values were provided but surface and bed elevation values existed, we simply calculated ice thickness by subtracting the surface value from the bed value. At times, bed elevation was higher than surface elevation, likely due to issues with the semi-automatic picker used to extract the surface and bed reflector or a lack of distinctive reflectors in areas of shallow ice. To prevent this affecting the gridded product, we converted these values to –9999 for both the surface and the bed. Finally, we also conducted routine checks on the ice thickness data by comparing the given ice thickness value with the inferred ice thickness calculated from subtracting surface with bed. If these did not match, we placed –9999 on the ice thickness values.

File naming conventions were also used throughout to easily identify a specific dataset as follows: DataProvider_Year_CampaignName_TypeofData_BM3. The type of data used was separated into three categories: Airborne Radar (AIR), Ground-based Radar (GRN), and Seismic (SEI) data. The "BM3" abbreviation at the end identifies the datasets was part of the Bedmap3 compilation to differentiate from the Bedmap 1 and 2 (BM1 and BM2) compilations. For instance, the file named 'NASA_2019_ICEBRIDGE_AIR_BM3.csv' refers to the ICEBRIDGE airborne campaign led by NASA in 2019.Providing an overall uncertainty value for all the bed elevations compiled by Bedmap is challenging due to the amount of data providers and radar systems used in the last 60 years (see Appendix tables). This uncertainty is often calculated as the RMS error of bed elevation values at crossover points across a survey area (e.g. Fremand et al., 2022). This error typically amounts to tens of meters and is constrained by changing bed characteristics, the radar system used and/or the processing of the data, as well as the value used for the propagation of radar waves through ice which is used to convert the radar two-way travel time to depth in meters. The metadata compiled by Bedmap for each survey provides information on whether any firn correction has been applied to the elevation values and on the value used for speed of electromagnetic waves through the ice.

In order to address the uncertainty in elevations values for the entire Bedmap dataset, we provide standard deviation, interquartile range, and standard error statistics parameters which are key to determine the variability of values in each 500mx500m pixel. The standard deviation represents the typical deviation of each data point to the mean value of the specific pixel and thus can be used to assess how accurately the mean value is representative of the real values. The standard error gives information about the variability across all the data points in the specific pixel and is used to estimate how well a specific data point is representative of the whole population. A high standard error indicates that the data within a specific pixel are widely spread around the population mean. The interquartile range calculates the difference between the first quartile and the third quartile and is used to measures the variability of the middle 50% of all values. Contrary to the standard error and standard



deviation, the interquartile range is not affected by extreme outliers that are present in a specific pixel. Together, these
parameters are used to assess the level of confidence in the data, where low values reflect a stronger fidelity in the data.

We also note that the spatial accuracy of datasets included in Bedmap2 and Bedmap3 is significantly higher than for
Bedmap1 due to the use of high-resolution GPS data, which have allowed for much better accuracy in the location of the
measurements for all surveys acquired from 1990s onwards. The accuracy of each bed elevation or ice thickness values can
vary from sub-meter accuracy for modern GPS measurements (Fremand et al., 2022) to several kilometres for data compiled
as part of the Bedmap1 dataset (Schroeder et al., 2019).

As part of the follow-up publication to this paper introducing the new Bedmap3 gridded products, we will include the
final grids and maps that will study and exclude possible cross-over errors and other possible problems in order to provide
high-quality gridded products.

## 4. Publishing the Bedmap source data

The Bedmap source data are available via the UK Polar Data Centre (PDC, https://www.bas.ac.uk/data/uk-pdc/, last
accessed: 29 July 2022), a trusted repository whose purpose is to manage polar datasets. Part of the Environment Data Services
(EDS) of the Natural Environment Reasearch Council (NERC) and certified by the CoreTrustSeal
(https://www.coretrustseal.org/, last access: 29 July 2022), PDC applies best data management practices and requirements to
facilitate reuse of the datasets stored on its data catalogue (https://data.bas.ac.uk/, last access: 29 July 2022). To increase the
discoverability of the datasets, a specific data portal – the SCAR BEDMAP data Portal (bedmap.scar.org, last access: 18
October 2022) has been developed by the Mapping And Geographic Information Centre (MAGIC) team from the British
Antarctic Survey. The publishing procedure described here is expected to be used for all upcoming versions of Bedmap with
regular updates allowing new source data to be easily accessible to the community.

Below, we discuss the release of the datasets centered around the FAIR data principles (Section 4.1), and present the data
portal infrastructure and its functionalities (Section 4.2).

### 4.1. FAIR data publishing

The source data for each version of Bedmap has been published as two separate Digital Object Identifier (DOI) datasets,
the first dataset contains all the standardised CSV files described in Section 3.1, the second dataset contains all the lines and
point shapefiles as discussed in Section 3.2.

The derived, gridded Bedmap products are also published as separate DOI datasets. Previously available through ftp
services, Bedmap2 products are now citable and properly stored for long-term preservation. Table 4 presents the different links
to the data for the different versions of Bedmap.



**Table 4. List of references for the Bedmap products.** For each Bedmap product (Bedmap1, Bedmap2 and Bedmap3), we provide
the link to the standardised CSV and shapefile data.

| Bedmap version | Standardised CSV | Shapefile points |
|---|---|---|
| **Bedmap1** | https://doi.org/jg6q (Lythe et al. 2022) | https://doi.org/jg6s (Lythe et al. 2022) |
| **Bedmap2** | https://doi.org/jg6r (Fretwell et al., 2022) | https://doi.org/jg6t (Fretwell et al., 2022) |
| **Bedmap3** | https://doi.org/jg6n (Fremand et al., 2022) | https://doi.org/jg8b (Fremand et al., 2022) |

In order to make the data findable, ISO 19115/19139 metadata are provided for each dataset. Each metadata record
provides general information about the dataset and is registered and indexed accordingly in the UK Polar Data Centre (PDC)
data catalogue Discovery Metadata System (https://data.bas.ac.uk/, last accessed: 29 July 2022) as well as the NERC data
catalogue (https://data-search.nerc.ac.uk/, last accessed: 29 July 2022). More detailed information can be found in the extended
header of the CSV data, such as the name of the data providers, the funding received, and a reference to cite the data. Althought
the metadata are succinct, it is possible to easily transform the CSV data to NetCDF with extended metadata as shown in the
Geophysics Book (https://antarctica.github.io/PDC_GeophysicsBook/BEDMAP/Get_full_metadata_from_CSV_file.html,
last accessed: 29 July 2022).

A DOI is provided for every Bedmap dataset (Table 4), making them easily retrievable and citable. For previously
published datasets, the original DOI is provided in the source metadata (Section 3.2) to ensure traceability and should be used
when the survey is used individually.

For universal accessibility, the data is downloadable through a standard HTTPS-protocol where no login account is
required. We used the web-based RAMADDA (Repository for Archiving and MAnaging Diverse DAta;
https://geodesystems.com/) data repository system which is an open-source content and data management platform and works
following a simple folder structure, with datasets organised by data-provider name to replicate the structure of the Data Portal.

To enhance interoperability and reusability, we published the underlying data using a specific CSV format, with detailed
and standardised variable names coming from FAIR vocabularies (see Table 3 and Section 3.2). To be re-usable, the data is
released under Creative Commons license CC-BY (https://creativecommons.org/licenses/by/4.0/, last accessed: 29 July 2022)
which allows any user to use the data freely and with flexibility, whilst at the same time ensuring full acknowledgment of those
involved in the collection and processing of the data. Keywords from the Global Change Master Directory (GCMD, 2021) are
used to describe the data in a consistent and comprehensive manner and increase the interoperability of the datasets. The end
goal is to provide all the information necessary for effective, long-term data re-use. In addition, we developed interactive,
open-source Jupyter Notebook tutorials written in Python to interact with the data programatically. Codes to convert the
standardised CSV files to the point and line files as described in section 3.3 are for example provided. These resources are

archived on the BAS GitHub repository and provided via an interactive web interface using Jupyter Book (https://antarctica.github.io/PDC_GeophysicsBook, last accessed: 29 July 2022). In addition, a specific python package allows
to read and plot the specific CSV formatted data (https://github.com/paul-breen/xcsv, Paul Breen, 2022, last access: 29 July 2022). We believe these to be particularly beneficial for assisting users in accessing the data and reproducing their own gridded products independently of the Bedmap project.

### 4.2. The SCAR Bedmap data portal

The newly-developed SCAR Bedmap Data Portal (bedmap.scar.org, last access: 18 October 2022) provides a common
endpoint for interacting with the Bedmap source data and products. The webmap architecture is based upon the well-used SCAR Antarctic Digital Databse webmap (https://www.add.scar.org/). The Data Portal is divided into five layer-menus: "Base layers", "Topographic information", "BEDMAP1", "BEDMAP2", and "BEDMAP3". The first menu contains the gridded products: ice bed, surface and thickness grids for Antarctica, currently contains Bedmap2 grids; these will be updated with new Bedamp3 grids as they become available. The second menu contains general topographic information such as coastline
and Ice-Land surface contours taken from the Antarctic Digital Database (https://www.bas.ac.uk/project/add/, last accessed: 29 July 2022). The three Bedmap tabs contain the shapefile layers for individual campaigns. When clicking on a point of the map, the user has direct access to the information of the survey and statistics for the specific point (see Table 3).

At the top of the interface several widgets are available, designed to help users with basic tasks such as measuring distances, areas and elevations or search for specific place names. The link to the direct download repository is also provided.

**5. Conclusion**

We have presented here the release of the source survey data on ice thickness, bed and surface elevation data used in Bedmap gridded products, including the upcoming Bedmap3. Altogether, this data release represents over 77 million data points collected as part of 270 campaigns since the 1950s. In addition to the previous Bedmap 1 and 2 datasets, we have here gathered new ice thickness data from 75 surveys, adding 50 million additional data points to those previous compilations. We
have developed a standardised CSV format in order to ensure interoperability between the different datasets, which we have checked following a specific quality control procedure and summarised on a 500mx500m grid to provide key statistics at the scale needed for the Bedmap3 gridded products.

The data have been published following the FAIR data principles. In particular, we have provided extensive metadata
with commonly-used keywords and have developed a data portal that provides a user-friendly interface to interact with and download the data. By providing and displaying both the source data and grids, the data portal allows any user to investigate the uncertainty of the gridding in specific areas and analyse differences between measurements and gridded interpolations.

We believe that this data release will considerably benefit the glaciology and broader Earth science community,
particularly in emerging fields such as machine learning and geostatiscs which can now make use of this wealth of fully
standardised data, as well as reproduce and create new compilation grids at different scales indenpendently from the Bedmap
grids. We believe that the standardisation and free availability of previously-unpublished datasets presented here will lead to
improved assessment of fundamental properties of the Antarctic Ice Sheet and predictions of its future contribution to sea level
rise, as well as  significantly increasing the life cycle of these important data.

**6. Data Availability**

All the data included in this manuscript are freely available from the UK Polar Data Centre (https://data.bas.ac.uk, last
access: 19 October 2022) and the SCAR Bedmap Data Portal (bedmap.scar.org, last access: 19 October 2022). Direct link to
the metadata and data can be found in Table 4. The data can be downloaded directly from the Ramadda interface by clicking
on the 'GET DATA' link from the metadata page or using wget commands as documented in the following instructions:
https://antarctica.github.io/PDC_GeophysicsBook/BEDMAP/Downloading_the_Bedmap_data.html (last access: 20
November 2022).

When using this data, please also cite the DOI citation provided in the source CSV metadata if this exists.

**7. Code Availability**

The user guide for the data portal and the Jupyter Notebook tutorials designed for reading the standardised CSV ice bed,
elevation and thickness data or create the shapefiles are accessible on the Jupyter Book interface under the BEDMAP3 section
(https://antarctica.github.io/PDC_GeophysicsBook, last accessed: 29 July 2022) or via the BAS GitHub repository
(https://github.com/antarctica/PDC_GeophysicsBook, last accessed: 29 July 2022). A specific library called xcsv
(https://github.com/paul-breen/xcsv, Paul Breen, 2022, last access: 29 July 2022) allows to read and plot data in the extended
CSV format as described in Section 3.2.

**8. Author contributions**

Alice Frémand., Peter Fretwell, and Julien Bodart co-led this data release. A. Frémand and J.  Bodart standardised
the data. The Jupyter Notebook was developed by A. Frémand. P. Fretwell and H. Pritchard initiated the collaboration and P.
Fretwell liaised with all the data providers. A. Frémand wrote the initial manuscript with input from P. Fretwell, J. Bodart and
H. Pritchard. P. Fretwell designed and populated the web map. Elena Field helped with the design of the web map.

Earth System
Science
Data

Aitken, Bo, Eisen, Fretwell, Gillet-Chaulet, Greenbaum, Lee, Matsuoka, Morlighem, Pattyn, Popov, Pritchard, Roberts, Schroeder, Siegert, Steinhage, Tinto, Xiangbin and Young were all members of the Bedmap3 SCAR Core Group and contributed to the overall project and standardization criteria.

P. Fretwell, H. D. Pritchard, J.L. Bamber, R. Bell, C. Bianchi, R.G. Bingham, D. D. Blankenship, D. Callens, G. Casassa, G. Catania, K. Christianson, H. Conway, H.F.J. Corr, X. Cui, D. Damaske, V. Damm, R. Drews, G. Eagles, O. Eisen,
H. Eisermann, F. Ferraccioli, R. Forsberg, S. Franke, S. Fujita, Y. Gim, V. Goel, P. Gogineni, J. Greenbaum, B. Hills, R.C.A.† Hindmarsh, P. Holmlund, N. Holschuh, J.W. Holt, A. Humbert, R.W. Jacobel, D. Jansen, A. Jenkins, W. Jokat, T. Jordan, E. King, J. Kohler, W. Krabill, K.A. Langley, J. Lee, G. Leitchenkov, C. Leuschen, B. Luyendyk, J MacGregor, E. MacKie, K. Matsuoka, M. Morlighem, J.† Mouginot, F.O. Nitsche, Y. Nogi, O.A. Nost, J. Paden, F. Pattyn, S.V. Popov, M. Riger-Kusk, E. Rignot, D.M. Rippin, A. Rivera, N. Ross, A. Ruppel, D.M. Schroeder, M.J. Siegert, A.M. Smith, D. Steinhage, M. Studinger,
B. Sun, I.† Tabacco, B.K. Tinto, S. Urbini, D. Vaughan, B.C. Welch, D.S. Wilson, D.A. Young, and A. Zirizzotti contributed to the data. All the authors commented and contributed to the final edits of the manuscript prior to publication.

## 9. Acknowledgements

We would like to dedicate this paper to the many scientists who have collected geophysical field data in harsh and extreme conditions over the Antarctic ice sheets over the last sixty years. Their commitment, dedication and drive has
populated this dataset and has advanced Polar science beyond measure.

Funding for the British Antarctic Survey staff has come from Natural Environment Research Council core funds. Funding for the data collection has come from many grants, institutions and projects. These are individually cited where appropriate in the metadata of the datasets.

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
