# Peer review of "Antarctic Bedmap data: FAIR sharing of 60 years of ice bed, surface and thickness data"

_Earth System Science Data, 2022_

## Author Comment (AC1)

Alice Fremand
British Antarctic Survey
CB3 0ET, UK
Email: almand@bas.ac.uk

Friday March 31st, 2023

**Re: [essd-2022-355 (author) - final response.] Antarctic Bedmap data: FAIR sharing of 60 years of ice bed, surface and thickness data by A. Fremand et al.**

Dear ESSD,

We would like to thank both reviewers for their very positive and insightful reviews of our manuscript, as well as the editorial team for handling the review process.

We are very pleased to see that both reviewers recognised the importance and usefulness of our dataset and manuscript, as well as the time invested in standardising and publishing the data following the FAIR principles. Both reviewers have provided us with some excellent comments, which have undoubtedly improved the quality of our manuscript.

In this response letter, we begin by addressing the comments from Reviewers #1, followed by those made by Reviewer #2. We have formatted the comments of each reviewer in italics and have highlighted our responses in red below each comment.

We look forward to hearing your decision and stand-by in the meantime with any queries you might have.

With best wishes,

Alice Fremand (on behalf of all co-authors)
* * *
**Response to reviewer #1**

*Frémand and her co-authors in their paper entitled "Antarctic BedMAP data: FAIR sharing of 60 years of ice bed, surface and thickness data" make available all the geophysical data (mainly by radio echo-sounding) acquired since the 1960s and crucial in our knowledge of the topography of the bedrock elevation under the Antarctic ice sheet. The opening of these data is an important advance for the community and was widely expected. It will allow various teams to take advantage of the initial dataset to allow the emergence of new methodologies and propose more accurate Digital Elevation*

*Models (DEMs) that are compatible with the physics of ice sheet models in particular (e.g. see the BedMachine initiative). These DEMs are an essential boundary condition for modeling ice dynamics and projecting the future of Antarctica and its contribution to sea level rise. I therefore consider this paper important, well structured and recommend it for publication after minor adjustments.*

We would like to thank Reviewer #1 for their positive comments on our manuscript and for their thorough review, which has improved this manuscript.

- *Web links appears with a date of access, I presume this is an editorial issue which I guess will be solved before the final publication?*

Thank you for your comment. The web links appears with a date of access as per the guideline of the journal. All dates of access have now been revised to comply with the date of revision.

- *On first reading, I had trouble understanding whether the shapefile data was the same as the CSV data. I admit that this is fairly obvious when you look at Tables 2 and 3, but the doubt lingered with me for a while. I think it comes from the introduction to section 3.3, the first sentence starting with "the CSV data were converted to shapefile" (L. 260). I think that presenting the need to offer summarised data might come before using the word "convert" which seems to me to be a bit of a misnomer, it is more than a format conversion.*

Thank you for your comment. To avoid any confusion in the future, we have changed the sentence to: "In addition to providing standardised CSV data (see Section 3.1), we also provide the data as shapefile and geopackage lines and statistically-summarised points." (Line 265)

- *L 262-264. "For Bedmpa1 (there is a typo), due to the difficulty... not possible to convert BedMap1... : only the bedmap1 shapefile points are provided". I don't understand this sentence, I see a contradiction, the conversion is not possible but the shapefile is provided. This needs to be rephrased. I presume the idea is that BEDMAP1 is provided as only one shapefile and not split by campaign but this is not clear.*

Thank you for your comment. To avoid any confusion, we have changed the sentence to: "For Bedmap1, due to the sparsity of points, it was not possible to convert the data to shapefile or geopackage lines, thus, only the Bedmap1 shapefile points are provided as part of this data release. Please note also that the Bedmap1 data are not split per campaign as per the Bedmap2 and Bedmap3, and is only provided as a single geopackage or shapefile point files." (Line 266)

- *Figure 2 panel d. Maybe the color scale could be adjusted, any red dot can be seen and it is hard to distinguish the two shade of orange used.*

Thank you for your comment. We agree with this point. To improve readability of the figure, the labels have been changed to only keep one shade of orange.

- *L 309. A space is missing after the dot.*

Thank you for your comment. This has been edited to add a space after the dot.

- *L 318. The word pixel is introduced when cell was used before. I believe that only one word should be used.*

Thank you for your comment. All mentions of the word 'pixel' has now been replaced by the word 'cell'.

- *Regarding the publishing of the data sources. Quantarcica ([https://www.npolar.no/quantarctica](https://www.npolar.no/quantarctica)) is widely used by the community, I would encourage the BEDMAP team to use this platform to further promote their dataset.*

This is indeed a good idea. We will liaise with the Quantarctica team to see if further collaborations are possible. We propose to do this after the publication of the Bedmap3 gridding products to offer all the products at once.

- *Regarding the access of the data. The front page is very clear ([https://www.bas.ac.uk/project/bedmap/#data](https://www.bas.ac.uk/project/bedmap/#data)), one could ask why the Bedmap1 gridding product is not provided. It becomes a bit more fuzzy when we visit the sub pages.*

Thank you for your comment. The Bedmap1 grid is now published and available through the front page. The citation is as follows: Lythe, M., Vaughan, D., & BEDMAP 1 Consortia. (2023). BEDMAP1 - Ice thickness, bed and surface elevation for Antarctica - gridding products (Version 1.0) [Data set]. NERC EDS UK Polar Data Centre. [https://doi.org/10.5285/908bb17f-467c-42bf-ae00-f03bb0feea23](https://doi.org/10.5285/908bb17f-467c-42bf-ae00-f03bb0feea23)

- *When we click on the links provided under the statistically-summarized data points Shapefiles, the title of the following pages have dropped the word « statistically summarized » to « standardized shapefiles ». This extends the confusion I mentioned earlier.*

Thank you for your comment. To avoid any confusion in the future, the main page ([https://www.bas.ac.uk/project/bedmap/#data](https://www.bas.ac.uk/project/bedmap/#data)) has been updated and the line/point data have been referenced under the 'Standardised shapefiles and geopackages – Lines and statistically-summarised data points' section. The metadata titles remain the same, only referencing 'standardised shapefiles and geopackages'.

- *When on one of the metadata or data webpages (e.g. [https://ramadda.data.bas.ac.uk/repository/entry/show?entryid=a72a50c6-a829-4e12-9f9a-5a683a1acc4a](https://ramadda.data.bas.ac.uk/repository/entry/show?entryid=a72a50c6-a829-4e12-9f9a-5a683a1acc4a)) there is a list of associated datasets. This list of associated datasets is not consistent from one page to another and do not understand why. If the aim is to promote the other datasets proposed by BEDMAP, I do not understand why these paragraphs are not simply entitled « Other BEDMAP products » with only one link to the front page.*

Thank you for your comment. The data pages have now been edited to reflect the change: they now reference the front page for the associated datasets.

- *I would find it useful if the data could be downloaded in a few clicks and not mission by mission (e.g. I would have liked to be able to download all the BedMap3 shapefiles in one click for example)*

Thank you for your comment. Although it is not possible to download all the Bedmap3 data in one go from the interface yet, we have developed a user guide ([https://antarctica.github.io/PDC_GeophysicsBook/BEDMAP/Downloading_the_Bedmap_data.html](https://antarctica.github.io/PDC_GeophysicsBook/BEDMAP/Downloading_the_Bedmap_data.html)) that explains how to download the data programmatically.

**Response to reviewer #2: Johnathan Kool**

*This is a significant paper, and one that will be of great interest to researchers. Not only does it mark the release of a significant data set, but it also does an excellent job of addressing data management considerations. The information is presented clearly, and there are many aspects which are highly citable.*

We would like to thank Reviewer #2, Johnathan Kool for his positive comments on our manuscript and for his thorough review, which has improved this manuscript.

*I have only minor suggestions and corrections, listed below.*
*Specific Comments:*
*Line 115 (suggestion): It may worth referencing Section III 1 c of the Antarctic Treaty here - identifying the need beyond a moral imperative to make the data available*

Thank you for your comment. The section III 1c of the Antarctic Treaty has been added as suggested: "Given the rapidity of change affecting large parts of the Antarctic Peninsula and threatening the stability of the West Antarctic Ice Sheet, and the urgency in predicting future ice loss (e.g. Mouginot et al., 2014; Golledge et al., 2015; DeConto & Pollard 2016; Gardner et al., 2018; Seroussi et al., 2020; Levermann et al., 2020), it is essential, beyond the legal imperative stated in the Section III 1 c of the Antarctic Treaty for these data to be freely available to the international community."

*Line 246 (suggestion): Consider providing an abbreviated example of what the header and lines might look like. Information can be condensed using '...', but I think an example would make it easier to visualise the content of this paragraph.*
Thank you for your comment. For clarity and following your suggestion, a figure showing an example of the header information has been added with its specific description in the caption (Figure 2). The information in the main text has also been condensed. The new sentence (Line 250) reads: "The format consists of (i) an extended header section, (ii) a header row composed of the column name following the CF convention and units in parentheses; and finally (iii) the data using comma as the separator."

*Line 294 (question): -9999 values often present problems for interpolation, or situations where data ranges can encompass -9999. Although that is not the case here - are there any alternatives for better representing NULL/NaN/Uknown values? I recognise that this is an issue with CSV, but it may be worth thinking about better ways of encoding this information.*

Thank you for your comment. The choice to use -9999 to highlight null values was driven by the community. This requirement was mainly chosen to be easily imported into software where NULL/NaN/Unknown values are not handled easily.

*Lines 309 & 326 (question) - You discuss how uncertainty in elevation values was handled, but was there also consideration for varying spatial accuracies?*
For data acquired from 1990s onwards, the spatial accuracy is higher than the cell size. It is thus only for older data (mainly Bedmap1) that the varying spatial accuracies may affect the quality of the overall grid, as we highlighted in lines 201-207 of the paper.
For Bedmap1, as all the underlying surveys have been standardised together, the statistics parameters also reflect the varying spatial accuracies. Indeed, in an area with strong elevation variability, if a point is not well positioned, there will be high discrepancies between the value points. The standard deviation will be high, highlighting a low level of confidence in the data.
To highlight this point, the following sentence has been added to the manuscript: "As this spatial uncertainty impacts the position of the elevation values and therefore their accuracy, the elevation

uncertainty statistics parameters can be used to indirectly assess the confidence in the spatial accuracy. However, the statistics parameters are only meaningful if a representative set of points are used to calculate the ice thickness, bed and surface elevation." (Line 328)

*Technical comments:*
*Figs 1&2 - Polar stereographic projection?*

The maps shown in Figure 1 and 2 (now identified as Figure 3) have indeed been mapped using the polar stereographic projection. The mention has been added to the maps.

*There are many instances where intensifiers(e.g. see*
*https://advice.writing.utoronto.ca/revising/wordiness/) are unnecessarily used.*
*97: "user-friendly" subjective assertion - consider omitting.*
*148: "rapidly" is there evidence or a citation?  Otherwise omit.*
*153: "considerable" - consider omitting*
*154:  "Indeed," – unnecessary*
*170: "in order" – unnecessary*
*265: "extremely" – unnecessary*
*267: "Such an" – unnecessary*
*356: "In order" – unnecessary*
*365: "easily" - subjective assertion - consider omitting*
*377: "full" – unnecessary*
*386: "particularly" unnecessary*
*Paragraph at 414 (several): Change to "This data release will benefit the glaciology and broader Earth science community, particularly in emerging fields such as machine learning and geostatistics which can now make use of this standardised data, and reproduce and create new compilation grids at different scales independently from the Bedmap grids.  These standardised, freely available, and previously-unpublished datasets will lead to improved assessments of fundamental properties of the Antarcitc Ice Sheet and predictions of its future contributions to sea level rise, increasing the (life cycle) of these important data."  -- ('life cycle' is an odd choice of words - maybe just 'value'?)*

Thank you for your comment and the explaining link. This is very helpful and will be useful for any future paper. All unnecessary intensifiers have been removed from the manuscript.

*Line 455 - Normally I leave Acknowledgements alone - but "beyond measure" seems a bit hyperbolic.  It's up to you.*

Thank you for your comment. The sentence has been edited as follows: "Their commitment, dedication and drive has populated this dataset and has greatly advanced Polar science." (Line 461)

*Typos:*
*Line 111: Theses - These*
*Line 361: Althought - Although*
*394: Bedamp3 -> Bedmap3*
*415: geostatiscs -> geostatistics*
*416: indenpendently -> independently*
*Line 422: Direct link -> Direct links*
*Line 423: Capitalise RAMADDA (or standardise usage)*
Thank you for highlighting the typos. They all have been corrected in the manuscript.

---

## Author Response (AR2)

Alice Fremand
British Antarctic Survey
CB3 0ET, UK
Email: almand@bas.ac.uk

Wednesday April 12th, 2023

**Re: [essd-2022-355 (author) - manuscript accepted with corrections] Antarctic Bedmap data: FAIR sharing of 60 years of ice bed, surface and thickness data by A. Fremand et al.**

Dear ESSD,

Thank you very much for accepting our paper to ESSD. Please find attached our revised manuscript following your constructive feedback.

In particular,  we have deleted the first sentence to avoid any confusion. The Figure 2 has now been transformed into a table and the subsection title of our Jupyter notebook has been changed to "Plotting flight lines" (see https://antarctica.github.io/PDC_GeophysicsBook/BEDMAP/Create_ShapeLines_for_BEDMAP3.html#plotting-flight-lines).

With best wishes,

Alice Fremand (on behalf of all co-authors)